# Estimating Residential and Industrial City Gas Demand Function in the Republic of Korea—A Kalman Filter Application

**Chansu Lim** [1,2]

1 R&D Center, GS Caltex Corp, Daejeon 34122, Korea; chancelim@gmail.com or chance@gscaltex.com;
  Tel.: +82-10-6535-2812
2 Program of Technology Management, Economics, and Policy, School of Engineering, Seoul National
  University, Seoul 151-742, Korea

**Abstract:** This paper analyzes the city gas demand function in Korea from 1998 to 2018. The demand function of city gas is derived by a Kalman filter method, and price and income elasticities varying with time are estimated. In the case of residential city gas, the price elasticity gradually decreased to a value of approximately 0.57, while income elasticity increased to approximately 1.48 from 1998 to 2018. Alternatively, industrial city gas demand's price and income elasticities have been estimated as inelastic, as their absolute values were less than unity over time. The absolute values of price and income elasticities are estimated to be larger for residential than industrial city gas, and thus, city gas consumers are more likely to respond to changes in price and income for residential than industrial city gas. There is a substantial income effect on demand for residential city gas in Korea, whereas industrial city gas is found to have relatively small income and price effects. The results of this study provide policy makers with a Kalman filter method to access more accurate information on the city gas demand function's elasticities, which change with time.

**Keywords:** city gas demand; Kalman filter; price and income elasticity

## 1. Introduction

City gas is an eco-friendly clean fuel that does not generate atmospheric pollutants—such as $SO_X$ (sulfur oxides), $NO_X$ (nitrogen oxides), and PM (particulate matters)—in comparison to such fossil fuels as coal and fuel oil. City gas primarily supplies homes, commercial buildings, and industrial complexes through pipelines, and is widely used for cooking and heating because of its convenience. City gas in Korea has already become a necessity good for the national economy, similar to electricity, water, and sewage services.

Korea's city gas demand increased by 18,967 million $m^3$ in 2013, and its demand then declined by 20,043 million $m^3$ in 2018 (Figure 1). On the one hand, Korea's city gas industry is experiencing a period of demand for expansion and supply stability, and its efficiency-based industrial structure is changing. On the other hand, current low oil prices continue (50–70 $/bbl), and the price competitiveness of city gas has decreased compared to such competitive energy sources as liquefied petroleum gas, kerosene, and electricity.

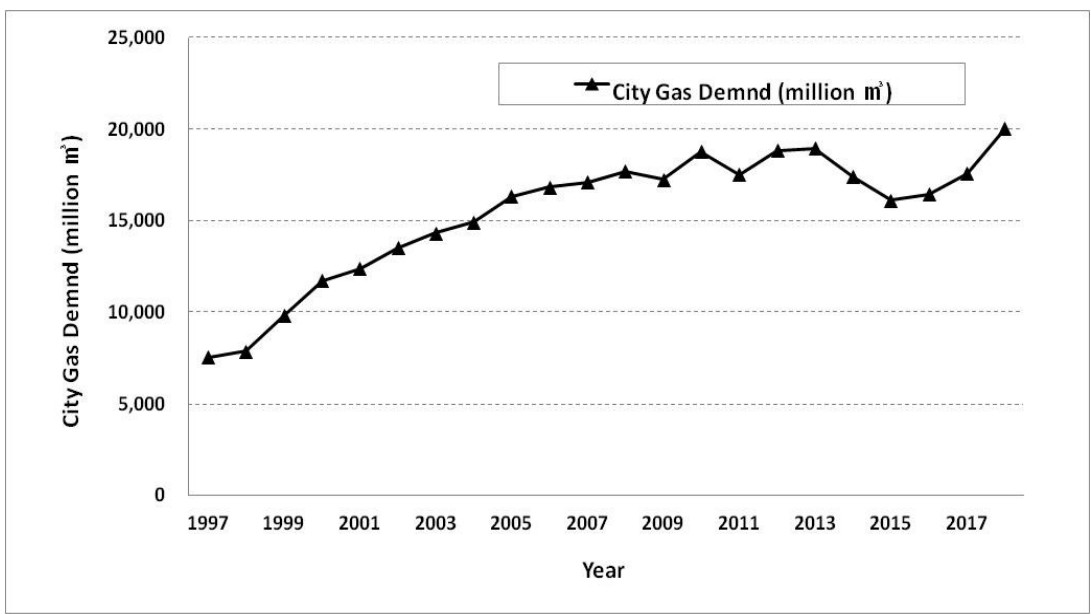

**Figure 1.** City gas demand in the Republic of Korea (Unit: million m$^3$) Source: Korea City Gas Association, Statistics of Korea City Gas Industry [1].

The accurate forecasting of city gas demand is fundamentally important as a basis for implementing reasonable energy policies. The elasticity of city gas demand indicates the sensitivity of demand changes with such explanatory variables as price and income. Thus, it is possible to provide information on city gas demand behaviors in each location where city gas is consumed.

This study estimates the demand function of the household and industrial sectors, which use the most city gas, to derive both price and income elasticities. Although the Kalman filter method is primarily used in control engineering, this is applied in the current study versus traditional econometric methods, such as time-series and panel data analyses, to quantitatively analyze the price and income elasticities which change over time.

The Kalman filter as first proposed by Rudolf E. Kalman is a recursive filter that tracks the state of linear dynamics, including noise [2]. The Kalman filter method has played an important role in the industrial electronics community and many engineering fields since the 1970s. A common application is for guidance, navigation, signal processing and the control of vehicles, particularly aircraft and spacecraft. [3] Furthermore, Kalman filters are also one of the main topics in the field of robotic motion planning and control, and they are sometimes included in trajectory optimization. The Kalman filter also works for modeling the central nervous system's control of movement. Due to the time delay between issuing motor commands and receiving sensory feedback, the use of the Kalman filter supports a realistic model for making estimates of the current state of the motor system and issuing updated commands [4].

A majority of analyses from previous studies regarding the forecasting of the city gas demand function are based on time-series and panel data. Balestra and Nerlove [5] estimated the city gas demand function from 1957 to 1962 using 36 cross-sectional panel datasets in the United States. Their results revealed price and income elasticities of -0.01 and 0.2, respectively, and both estimates were inelastic. Dahl [6] estimated the relative price and income elasticities of electricity and city gas demand; on average, the price and income elasticities were -0.27 and 0.71, respectively. Lee and Singh [7] used cross-sectional panel data for the US state of California to estimate long-term income elasticity for residential city gas, and found that long-term income elasticity is inelastic. Asche, Nilsen, and Tveterås [8] also used a panel data model, and found that the absolute values of long- and short-term income elasticity were less than one, indicating that both were inelastic. Aras and Aras [9] estimated the city gas demand function based on monthly time-series data of city gas consumption

in Turkey from 1996 to 2001 and temperature data from this period. In this case, the heating and non-heating periods were predicted separately. Bilgili [10] used a panel-DOLS (dynamic ordinary least squares) method for OECD countries' panel data—such as natural gas consumption, price, and income—from 1976 to 2006 to estimate price and income elasticities as -1.292 and 1.032, respectively. Yu et al. [11] estimated the price and income elasticities of residential demand for natural gas using panel data for Chinese cities during the period of 2006–2009, and found that natural gas consumption is price elastic and income inelastic when other covariates (e.g., the supply of a natural gas pipeline and heating degree days) are controlled. Zhang et al. [12] constructed an autoregressive distribution lag model to study the elasticity of natural gas demand in various sectors of China, and found that the long-term price elasticity of natural gas demand is larger than 0.

In recent years, many studies have predicted the demand function of city or natural gas through models other than those in existing econometric methods. Zhu et al. [13] applied a support vector regression-based forecasting method to daily natural gas demand forecasting in the United Kingdom that demonstrated better prediction results than the autoregressive moving average model. Ozdemir et al. [14] constructed a natural gas consumption function based on hybrid genetic algorithm-simulated annealing that incorporated Turkey's natural gas consumption, GDP, population, and economic growth rates from 2001 to 2009. Azadeh et al. [15] analyzed the daily natural gas demand in Iran during the period from 22 December 2007 to 30 June 2008, using an adaptive network-based fuzzy inference system (ANFIS), and showed that ANFIS provided better fitted results than artificial neural network (ANN) and conventional time series approaches.

This study contributes to the existing research on energy policy, economy and engineering in at least two respects. First, although there are numerous studies on the demand function for city gas (or natural gas) in other countries, there are much fewer studies on Korea's city gas demand. To the best of my knowledge, few studies have analyzed the elasticities of Korea's city gas demand function. Yoo et al. [16] estimated the price and income elasticities of residential city gas in Seoul, Korea using the bivariate specification of sample selection model and 380 household survey data points. It was found that the price elasticity was inelastic as -0.243 and the income elasticity was also inelastic as 0.335, and provided insightful information for understanding Korea's city gas market. However, this study aims to give more information on Korea's city gas market by analyzing household (or residential) city gas as well as industrial city gas. Second, this study uses the Kalman filter method to analyze the city gas market. Few studies have analyzed the city gas demand function using the Kalman filter method to date; most studies have analyzed the elasticities of city gas demand using a traditional econometric model, such as time series and panel data analysis. As examples of applying the Kalman filter method to energy economics, Inglesi-Lotz [17] estimated the price elasticity of electricity in South Africa during the period 1980–2005, and Arisoy and Ozturk [18] estimated price and income elasticities in the electricity sector based on Turkey's electricity consumption from 1960 to 2008.

Thus, it is meaningful that this study predicts Korea's city gas demand function by applying the Kalman filter method instead of a traditional econometric method to estimate price elasticity and demand elasticity, as the former is primarily used in engineering.

## 2. Materials and Methods

### 2.1. Data

This study estimated the demand function of Korea's domestic and industrial city gas from January 1998 to July 2016 using its residential and industrial city gas consumption, price, and residential and industrial income using the Kalman filter method (Figure 2).

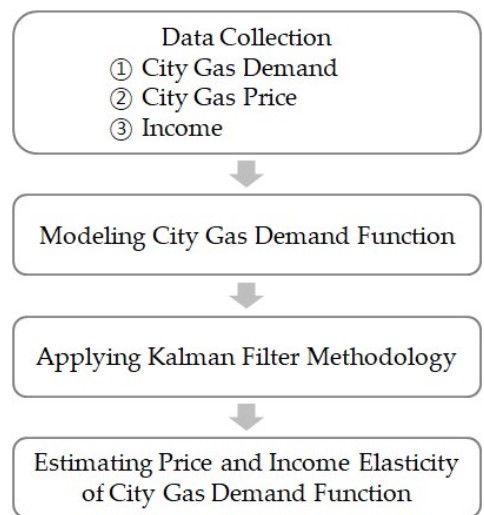

**Figure 2.** Overview of estimating city gas demand function.

The consumption of residential and industrial city gas (in $m^3$) and price data (in Korean won, or KRW) were prepared using the Korean Energy Statistical Information System [19]. Income data from the residential and industrial sectors during the same period (January 1998 to December 2018) were classified through the Korean Statistical Information System. Then, the average monthly urban household income account (excluding 1 person and farm) was classified as residential income, and manufacturing production in GDP was classified as industrial income [20].

Tables 1 and 2 display the descriptive statistics for residential and industrial city gas. On the one hand, the average consumption of residential city gas is approximately 1.73 times that of industrial city gas, and the average selling price of residential city gas is approximately 1.16 times higher than that for industrial use. On the other hand, the average residential income as average monthly urban household income is around 3.4 million KRW, and the average industrial income as manufacturing production is around 25 trillion KRW.

**Table 1.** Descriptive statistics of residential city gas in the Republic of Korea. KRW: Korean won.

| Descriptive Statistics | Residential City Gas Demand (in Million $m^3$) | Residential City Gas Price (in KRW/ $m^3$) | Residential Income (in KRW) |
|---|---|---|---|
| Mean | 835.06 | 618.47 | 3,409,389.95 |
| Std. | 560.67 | 177.90 | 871,328.88 |
| Max. | 2,158.00 | 973.61 | 5,034,564.00 |
| Min. | 167.00 | 334.42 | 1,893,747.00 |

**Table 2.** Descriptive Statistics of Industrial City Gas in the Republic of Korea.

| Descriptive Statistics | Industrial City Gas Demand (in Million $m^3$) | Industrial City Gas Price (in KRW/ $m^3$) | Industrial Income (in Billion KRW) |
|---|---|---|---|
| Mean | 481.50 | 531.02 | 25,701.57 |
| Std. | 206.10 | 193.62 | 8,306.55 |
| Max. | 1,004.40 | 917.59 | 39,330.27 |
| Min. | 113.80 | 240.65 | 10,693.70 |

To apply and construct the city gas demand function, all data regarding city gas demand, price and income were converted to logarithmic scale data. In general, the demand function is constituted by a logarithmic form to calculate the price and income elasticity easily [21].

*2.2. Methodology*

This study established the city gas demand function to estimate the city gas price and income elasticities using the Kalman filter method. It is an iterative estimation method to estimate and predict an unknown state using given information, and update this prediction through additional information, which will be reflected in a subsequent prediction. The Kalman filter keeps track of the estimated state of the system and the variance or uncertainty of the estimate. The estimate is updated using a state transition model and measurements.

An estimation method using the Kalman filter must first establish a state space model, which is expressed by measurement (or observation) equations and state (or transition) equations. Following Durbin and Koopman [22], the general state-space model using the Kalman filter method is presented as follows:

$$\text{Measurement Equation} \quad Y_t = X_t\beta_t + \varepsilon_t \quad \varepsilon_t \ : \ i.i.d. \quad N(0,R) \tag{1}$$

$$\text{State Equation} \quad \beta_t = \Gamma\beta_{t-1} + \nu_t \quad \nu_t \ : \ i.i.d. \quad N(0,Q) \tag{2}$$

The measurement equations represent the relationship between the observed and predicted variables at time t; the state equations are relational formulas of the predicted variables at times t and $(t-1)$; $Y_t$ denotes a measurement variable ($m \times 1$ column vector); $\beta_t$ denotes the state variable ($n \times 1$ column vector) to be predicted; and $X_t$ denotes an explanatory variable ($m \times n$ vector) mediating the measurement variable $Y_t$ and the state variable $\beta_t$. Further, $\Gamma$ is a parameter matrix describing the state equation and is assumed to be a unit matrix ($n \times n$) in a Kalman filter-applied model; and $\Gamma$ varies according to the error term $\nu_t$ with the covariance matrix $Q \ (n \times n)$, where $\varepsilon_t$ has the covariance matrix $R \ (m \times m)$.

The state variable $\beta_t$ at time t is estimated as $\widehat{\beta}_t^{-}$ using the estimated value $\widehat{\beta}_{t-1}$ at time (t - 1), as follows:

$$\widehat{\beta}_t^{-} = \Gamma\widehat{\beta}_{t-1} \tag{3}$$

$$P_t^{-} = Var(\widehat{\beta}_t^{-}) = \Gamma Var(\widehat{\beta}_{t-1})\Gamma^T + Q = \Gamma P_{t-1}\Gamma^T + Q \tag{4}$$

In this case, the matrix $P_t^{-}$ corresponds to the variance of $\widehat{\beta}_t^{-}$, and if $Y_t$ is measured at time t, the Kalman gain $K_t$ can be calculated through the conditional prediction error $\eta_t$ and the prediction error's covariance matrix $F_t$, as follows:

$$\eta_t = Y_t - X_t\widehat{\beta}_t^{-} \tag{5}$$

$$F_t = Var(\eta_t) = Var(Y_t - X_t\widehat{\beta}_t^{-}) = X_tP_t^{-}X_t^T + R \tag{6}$$

$$K_t = P_t^{-}X_t^T F_t = P_t^{-}X_t^T(X_tP_t^{-}X_t^T + R)^{-1} \tag{7}$$

The state variable's estimation value $\widehat{\beta}_t$ can be obtained through the following calculated Kalman gain $K_t$, and the covariance $P_t$ of $\widehat{\beta}_t$ can be calculated as follows:

$$\widehat{\beta}_t = \widehat{\beta}_t^{-} + K_t(Y_t - X_t\widehat{\beta}_t^{-}) \tag{8}$$

$$P_t = P_t^{-} - K_tX_tP_t^{-} \tag{9}$$

Through $\widehat{\beta}_t$ and the covariance $P_t$, $\widehat{\beta}^-_{t+1}$ and $P^-_{t+1}$ can be calculated as follows, with $\widehat{\beta}_{t+1}$, the covariance $P_{t+1}$ and the Kalman gain $K_{t+1}$ at time (t + 1) derived using a recursive calculation method:

$$\widehat{\beta}^-_{t+1} = \Gamma \widehat{\beta}_t \tag{10}$$

$$P^-_{t+1} = Var(\widehat{\beta}^-_{t+1}) = \Gamma Var(\widehat{\beta}_t)\Gamma^T + Q = \Gamma P_t \Gamma^T + Q \tag{11}$$

$$\eta_{t+1} = Y_{t+1} - X_{t+1}\widehat{\beta}^-_{t+1} \tag{12}$$

$$F_{t+1} = Var(\eta_{t+1}) = Var(Y_{t+1} - X_{t+1}\widehat{\beta}^-_{t+1}) = X_{t+1}P^-_{t+1}X^T_{t+1} + R \tag{13}$$

$$K_{t+1} = P^-_{t+1}X^T_{t+1}F_{t+1} = P^-_{t+1}X^T_{t+1}(X_{t+1}P^-_{t+1}X^T_{t+1} + R)^{-1} \tag{14}$$

$$\widehat{\beta}_{t+1} = \widehat{\beta}^-_{t+1} + K_{t+1}(Y_{t+1} - X_{t+1}\widehat{\beta}^-_{t+1}) \tag{15}$$

$$P_{t+1} = P^-_{t+1} - K_{t+1}X_{t+1}P^-_{t+1} \tag{16}$$

This study applied the Kalman filter method to forecast the city gas demand function. Consequently, the price and income elasticities that vary with time can be estimated. The model based on the Kalman filter method is advantageous, as the city gas demand function can be quantitatively analyzed relative to the dynamic change according to the sales price of city gas as well as each entity's economic activity.

The demand function of residential and industrial city gas is generally expressed as a logarithmic function of price and income, as follows:

$$\ln Q_t = \beta_{0,t} + \beta_{1,t} \ln P_t + \beta_{2,t} \ln Y_t + \varepsilon_t \tag{17}$$

$$\begin{bmatrix} \ln Q_t & 0 & 0 \end{bmatrix} = \begin{bmatrix} 1 & 0 & 0 \\ 0 & \ln P_t & 0 \\ 0 & 0 & \ln Y_t \end{bmatrix} \times \begin{bmatrix} \beta_{0,t} \\ \beta_{1,t} \\ \beta_{2,t} \end{bmatrix} + \begin{bmatrix} \varepsilon_t & 0 & 0 \end{bmatrix} \tag{18}$$

where $Q_t$ is the demand ($m^3$) of city gas at time t, $P_t$ is the price (KRW/$m^3$), $Y_t$ indicates income (GDP, KRW), and $\varepsilon_t$ is the error term. Further, $(\beta_{1,\,t},\ \beta_{2,\,t})$ is an indicator of the degrees of influence of price and income, which are measured as price and income elasticities, respectively, they are set as demand functions in a logarithmic form.

Assuming that $\beta_t$ is the next state variable to be predicted and follows a random walk process, it can be expressed as $\beta_t = \beta_{t-1} + e_t$. These assumptions allow the parameter $\beta_t$, which is an estimated parameter and the price of demand for city gas, to change over time. Therefore, the equation for the state variable $\beta_t$ of the time-varying parameter (TVP) model in the residential and industrial city gas demand function is as follows:

$$\begin{bmatrix} \beta_{0,t} \\ \beta_{1,t} \\ \beta_{2,t} \end{bmatrix} = \begin{bmatrix} 1 & 0 & 0 \\ 0 & 1 & 0 \\ 0 & 0 & 1 \end{bmatrix} \times \begin{bmatrix} \beta_{0,t-1} \\ \beta_{1,t-1} \\ \beta_{2,t-2} \end{bmatrix} + \begin{bmatrix} e_t \\ e_{1t} \\ e_{2t} \end{bmatrix} \tag{19}$$

Therefore, this study's price and income elasticity of the city gas demand function can be estimated over time through measurement equations (Equations (1) and (18)), state equations (Equations (2) and (19)), Kalman gain (Equation (7)) and predicting equations (Equations (8) and (9)). The city gas demand function is composed of measurement equations, and such estimated coefficients as the price and demand elasticities are constructed by the state equation. Thus, the estimated coefficient's change over time is measured using the Kalman filter method; this is also called the TVP model [17]. As the TVP estimation method does not require stationary characteristics for the city gas data, it is

possible to measure the time-series data's structural change by allowing a time-dependent change in the state variable.

In this study, the price and income elasticity of the city gas demand function were estimated by using the statistical software R with package dlm [23], which was available from the Comprehensive R Archive Network at http://CRAN.R-project.org/package=dlm.

Therefore, this study does not assume static or fixed price and demand elasticities, which are primarily used in existing econometric studies. Consequently, it is meaningful to estimate the demand function, or price and demand elasticities, which can change in accordance with industrial developments and standards of living.

## 3. Results

The residential and industrial city gas demand functions were constructed by examining residential and industrial city gas consumption from January 1998 to December 2018, as well as the income from residential and industrial sectors. Each price and demand elasticity was then respectively estimated through the TVP model using the Kalman filter method.

First, in the case of residential city gas demand, the income elasticity was measured as lower than price elasticity from 1998 to 2006, but the income elasticity increased to be greater than price elasticity since 2006 (Figures 3 and 4). The income elasticity increased to approximately 1.48, which was greater than unity. On the other hand, the price elasticity was greater than income elasticity from 1998 to 2006, but the price elasticity decreased to be lower than income elasticity since 2006, and the value of price elasticity decreased to approximated 0.57. In other words, changes in price greatly affect city gas consumption rather than changes in income from 1998 to 2006, and conversely, changes in income have greatly affected city gas consumption rather than changes in price since 2006. In 2006, there were the crossover changes between price and income elasticity at the value of unity. Therefore, since 2006, the price elasticities for the residential city gas demand function have been inelastic to a value of less than one, but income elasticities are elastic with values greater than 1.

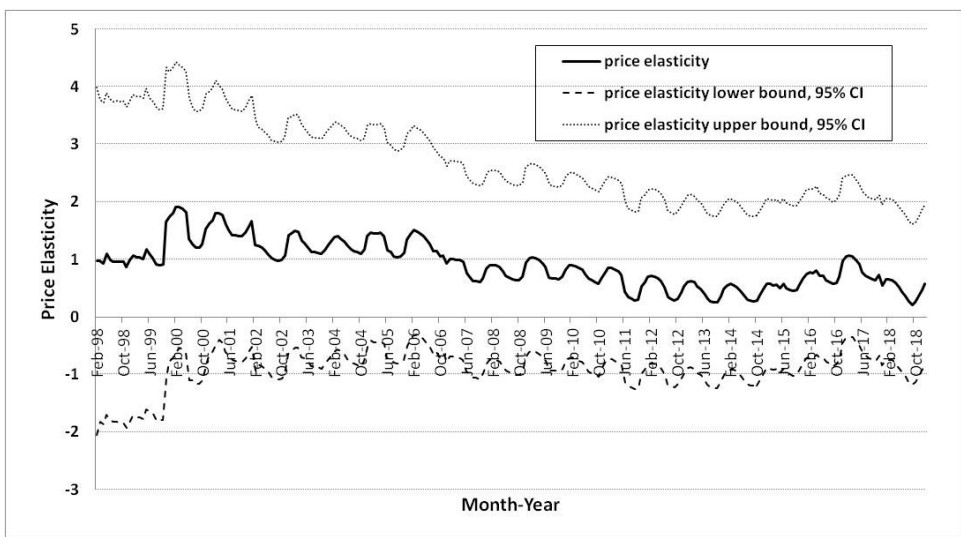

**Figure 3.** Residential city gas price elasticity estimates.

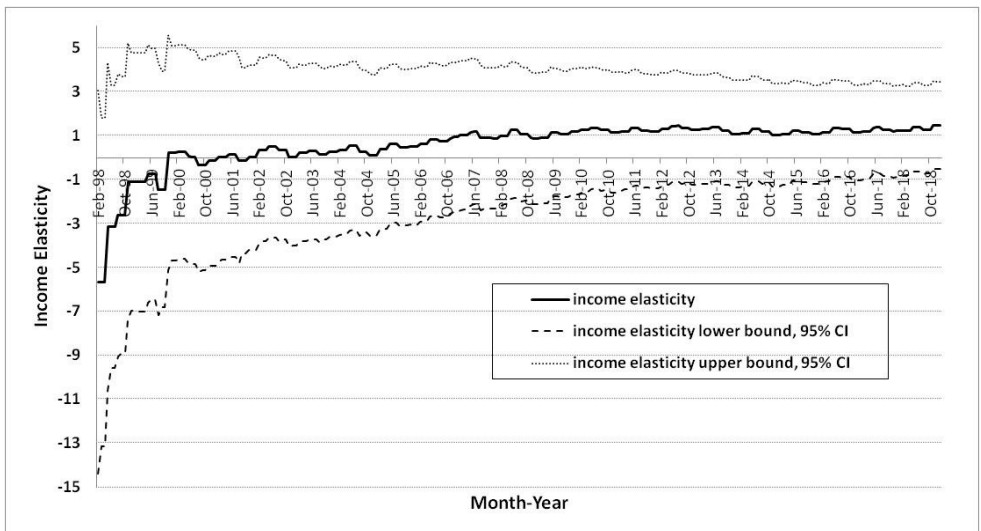

**Figure 4.** Residential city gas income elasticity estimates.

Therefore, residential city gas consumers have sensitively responded to changes in income, and this trend has been increasing in Korea since 2006. However, their responses have been impervious to price changes, and this trend has gradually decreased. Residential city gas consumption has been sensitive to changes in income, which is one reason for the supply of city gas pipelines. Korean residential income has increased since 1998, which has led city gas companies to expand their domestic city gas infrastructure (Figure 4). Thus, the expanded city gas infrastructures for households have increased the demand for residential city gas.

In the case of industrial city gas demand, the price elasticity is estimated to be larger than income elasticity, but the difference is very small (Figures 5 and 6). The price and income elasticities for industrial city gas in 1998 were 0.33 and 0.10, respectively. Further, the price elasticity has been negative since 2011, and income elasticity has been mostly negative since 1998 (excluding 1999), with 2018 values of -0.07 and -0.21, respectively.

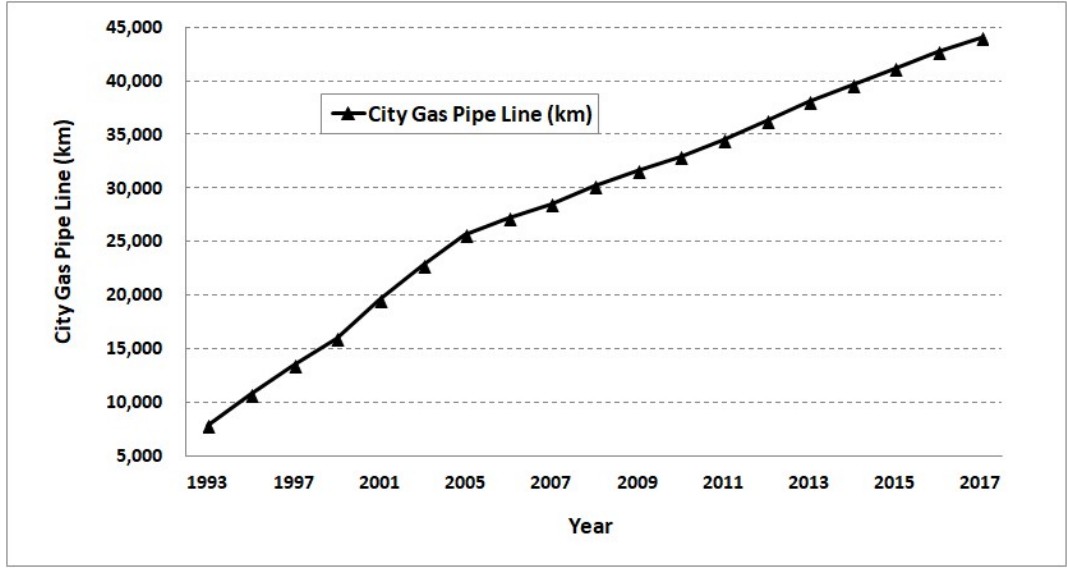

**Figure 5.** Total city gas pipelines in Korea (km). Source: Korea City Gas Association, Statistics of Korea City Gas Industry [1].

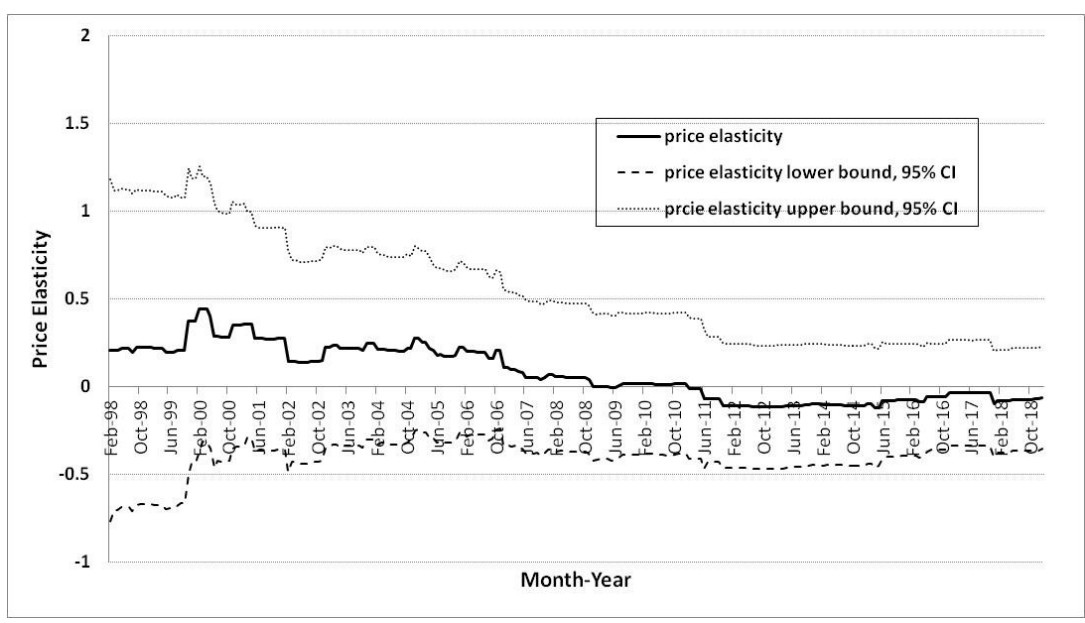

**Figure 6.** Industrial city gas price elasticity estimates.

The price and income elasticities for industrial city gas demands fluctuate at approximately zero. With values between -0.28 and 0.10, this indicates that industrial city gas does not change significantly with changes in prices and income, unlike residential city gas. This indicates that industrial city gas is used in factories' heat sources or industrial raw materials, and will not easily change due to changes in price and income. Thus, the demand for industrial city gas is inelastic, with values of less than one for both price and income elasticities (Figure 7).

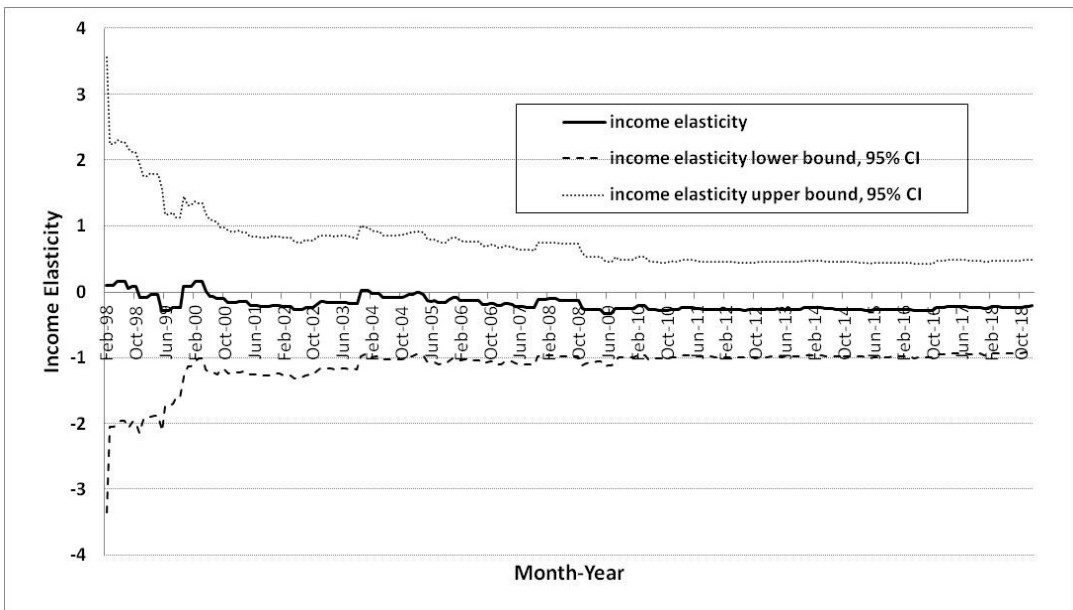

**Figure 7.** Industrial City Gas Income Elasticity Estimates

The industrial city gas demand's income elasticity has been mostly negative since 1998. As industrial income has increased, the demand for industrial city gas has decreased rather than increased. In other words, the demand for industrial city gas has been substituted by other competitive energy sources, such as liquefied petroleum gas, kerosene, and electricity. This electrification is one reason why the demand for industrial city gas did not increase. Korea is one of the OECD member nations with the largest electricity consumption among its industrial energy use (Table 3).

**Table 3.** Industrial electricity consumption share (2011).

| Countries | Value | Rank |
|---|---|---|
| Iceland | 86.6 % | 1 |
| Republic of Korea | 52.3 % | 4 |
| United States | 23.7 % | 34 |
| OECD Average | 32.8 % | - |
| OECD Average (without the Republic of Korea) | 31.7 % | - |

Source: IEA (International Energy Agency), World Energy Statistics and Balances (2011) [24].

Comparing residential and industrial city gas demands reveals that the absolute values of price and income elasticity are larger for residential than industrial city gas. Further, city gas consumers are more likely to respond to changes in price and income for residential than industrial city gas. The difference between price and income elasticities is larger for residential than industrial city gas, and thus, the demand for residential city gas is more affected by changes in income than demand for industrial city gas.

Table 4 showed the evaluation of the values of forecasting performance of the city gas demand function's time-varying elasticity estimates using the Kalman filter method, in the case of residential and industrial city gas, respectively. The evaluation of the values showed that the proposed models for the city gas demand function's elasticity estimates were well fitted with the data.

**Table 4.** Evaluating the forecasting performance of the city gas demand function's time-varying elasticity estimates using the Kalman filter method.

| Cases | MSE (Mean Square Error) | MSE (Mean Square Error) | MAPE (Mean Absolute Percentage Error) |
|---|---|---|---|
| Residential City Gas | 0.9879 | 0.8287 | 0.0942 |
| Industrial City Gas | 0.9883 | 0.8006 | 0.0928 |

## 4. Discussion

This study estimated the demand function of Korea's city gas as derived by a Kalman filter method, and price and income elasticities varying with time in both the cases of residential and industrial city gas. Because there have been few studies regarding Korea's city gas industry and demand, this study gives more details about Korea's city gas market.

There is a substantial income effect on demand for residential city gas in Korea, whereas industrial city gas is found to have relatively small income and price effects. The income elasticity of residential city gas increased to 1.48, which is different to previous studies [4–6,15]. The previous studies revealed that the income elasticity of overall natural gas was less than unity, which means it is inelastic in terms of income. However, this study shows that in the case of residential city gas, not overall natural gas level, income elasticity is not inelastic but elastic. In the other hand, the price elasticity of industrial gas was inelastic due to having absolute values of less than unity over time, which followed previous studies [4–6,15]. In terms of the price elasticity, residential city gas is more elastic than industrial city gas, which is an also more detailed analysis result of the Korean city gas market than previously. Comparing other countries and industries, the price elasticity for Turkey's electricity market was inelastic in terms of both residential and industrial electricity [17]. Additionally, the crossover changes between price and income elasticity occurred at the value of unity, which was an interesting event and not found in previous studies [4–6,15]. Korea's residential city gas market has experienced income effects with increasing demand for quality of life, clean energy and convenient usages.

This study provides policy makers with a Kalman filter method to access more accurate information regarding the city gas demand function's elasticities, which change with time. Previous studies analyzed city gas (or natural gas) demand using traditional econometric models,

such as time series and panel data models [4–6]. They gave static results for price and income elasticity which did not vary with time. However, this study shows the time-varying elasticity of price and income in city gas; thus, this study's results are helpful for policy makers to predict the future trends of price and income elasticity for city gas. In terms of industrial city gas, because of its small price elasticity and income elasticity, the demand for industrial city gas changes less with changes in prices and income. In terms of residential city gas, because of the relatively high price elasticity, the demand for residential city gas is likely to increase significantly if household gas price falls.

## 5. Conclusions

The Republic of Korea is currently facing challenges in not only effectively overcoming various social conflicts caused by a transitioning energy policy, but also promoting national security as well as an environmentally friendly energy policy. As city gas is an environmentally friendly clean energy resource, it can bridge the energy supply during the transition from fossil fuels to renewable energy. Therefore, the currently stagnant demand for city gas is likely to increase in the future.

This study derives the city gas demand function using a Kalman filter method, then estimates both price and income elasticity as they vary with time. In the case of residential city gas, the price elasticity has gradually decreased to a value of approximately 0.57, while income elasticity increased to approximately 1.48 from 1998 to 2018. In addition, there were crossover changes between price and income elasticity at the value of unity in 2006. Therefore, the price elasticity was inelastic to a value of less than one, but income elasticity was elastic with values greater than 1.

In the case of industrial city gas, price and income elasticities have been estimated as inelastic due to having absolute values of less than one over time. Specifically, the industrial city gas demand's income elasticity was negative, and thus, the demand did not increase as industrial income increased. Thus, other competitive energy sources provided a substitution to meet the demand for industrial city gas.

The absolute values of price and income elasticities are both larger for residential than for industrial city gas, and thus, the demand for residential city gas increasingly fluctuates with changes in price and income.

City gas has expanded its influence as a major energy source in the Republic of Korea, with the city gas industry increasing its need for a stable supply, demand management, decreased costs, and appropriate facility investments. To establish a reasonable energy policy, it is important to accurately and objectively predict the city gas demand function.

**Funding:** This research received no external funding.

**Conflicts of Interest:** The authors declare no conflict of interest.

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
