# Peer review of "Estimating Residential and Industrial City Gas Demand Function in the Republic of Korea—A Kalman Filter Application"

_sustainability, doi:10.3390/su11051363_

Round 1

Reviewer 1 Report

This paper estimates the demand function of residential and industrial sector using Kalman filter. It is well organized and written, but following comments should be revised for publication.

(Major comments)

1. In this study, author uses the data of private consumption and facility and construction investments in GDP for the residential income and industrial income respectively. Author can obtain the data of residential income from Korean Statistics System directly. Moreover, in general, energy consumption in industrial sector is affected by the total production not investment. Re-estimation (with income and total production) or backup (explanation for this data) is required.

2. Implication is weak. Therefore, estimation results need some comparison with other countries’ case or results of previous literature. Readers don’t know whether the estimated price and income elasticities of Korea are higher or lower than those of other countries. So, comparison with other countries is necessary for more discussion and implications.

3. What is the advantage of using Kalman Filter instead of traditional econometrics? Please explain the reason in manuscript.

(Some minor comments)

4. Line 27. ‘Public good’ should be changed to ‘necessity good’ or ‘necessary good.’ Theoretically, utilities such as electricity and gas are not public good.

5. Line30-33. All information is shown in Figure 1. Please delete numbers and just explain the trends based on Figure 1.

6. Line 87-92. I am not sure whether author uses the data of price, income and GDP which are adjusted reflecting inflation.

7. Line 154. Please check equation (16), which is not the vector form of equation (15).

8. It would be better to include ‘function’ in title. ‘Estimating demand’ is wrong expression and ‘estimating demand function’ is right one.

Author Response

Reply to the Review Report (Reviewer 1)

All mentioned comments are reflected in the attached revised paper with my best.

Please consider the positive and kind review for my revised paper.

Comment

Reply

1.     In this study, author uses the data of private consumption and   facility and construction investments in GDP for the residential income and   industrial income respectively. Author can obtain the data of residential   income from Korean Statistics System directly. Moreover, in general, energy   consumption in industrial sector is affected by the total production not   investment. Re-estimation (with income and total production) or backup   (explanation for this data) is required.

ü According to the comment, residential   and industrial income data were changed.

ü Period for data were extended   (1997-2018).

ü Re-estimation was done using the   changed data.

2.     Implication is weak. Therefore, estimation results need some   comparison with other countries’ case or results of previous literature.   Readers don’t know whether the estimated price and income elasticities of   Korea are higher or lower than those of other countries. So, comparison with   other countries is necessary for more discussion and implications

ü According to the comment,   estimation results were compared with other countries, such like United   States and China. And also compared with other industry like electricity   industry.

3.     What is the advantage of using Kalman Filter instead of traditional   econometrics? Please explain the reason in manuscript.

ü Additional literature studies   for Kalman filter method and applications were done in revised paper.

4.     Line 27. ‘Public good’ should be changed to ‘necessity good’ or   ‘necessary good.’ Theoretically, utilities such as electricity and gas are   not public good.

ü Changed to “necessity good” in   revised paper.

5.     Line30-33. All information is shown in Figure 1. Please delete   numbers and just explain the trends based on Figure 1.

ü Deleted detail numbers in   revised paper.

6.     Line 87-92. I am not sure whether author uses the data of price,   income and GDP which are adjusted reflecting inflation.

ü Re-estimation was done using data   with adjusted reflecting inflation in revised paper.

7.     Line 154. Please check equation (16), which is not the vector form of   equation (15).

ü Changed to vector form in   revised paper.

8.     It would be better to include ‘function’ in title. ‘Estimating   demand’ is wrong expression and ‘estimating demand function’ is right one.

ü Changed to ‘Estimating demand   function’ in revised paper.

Reviewer 2 Report

The authors have applied a recent model for their study. It’s very nice to see that the models are working nicely. In order to increase the value of the study, the authors may consider the following suggestions:

1.  The models are nicely described in terms of residential and industrial gas demand while differentiating the time varying income and price elasticity’s. The author may include some measures for evaluating the forecasting performance (For example, MAPE, or MSE). This is also essential to evaluate the fitness of the proposed models.

2.  To observe the behavior of the time varying beta coefficients, the author could also show the respective 95% CI for both income and price elascitity’s.

3. Please provide reference/s in Line 47 ( .... Kalman filter method is primarily used in control engineering .... ).

Thank you.  

Author Response

Reply to the Review Report (Reviewer 2)

All mentioned comments are reflected in the attached revised paper with my best.

Please consider the positive and kind review for my revised paper.

Comment

Reply

1.     The models are nicely described in terms of residential and   industrial gas demand while differentiating the time varying income and price   elasticity’s. The author may include some measures for evaluating the   forecasting performance (For example, MAPE, or MSE). This is also essential   to evaluate the fitness of the proposed models.

ü According to the comment, MSE(Mean   Square Error), MAD(Mean Absolute Deviation), and MAPE(Mean Absolute   Percentage Error) were calculated in revised paper

2.     To observe the behavior of the time varying beta coefficients, the   author could also show the respective 95% CI for both income and price   elascitity’s.

ü New Graphs with the respective   95% CI were inserted in revised paper.

3.     Please provide reference/s in Line 47 (Kalman filter method is   primarily used in control engineering).

ü Additional literature studies   for Kalman filter method and applications were done in revised paper.

Reviewer 3 Report

I have reviewed the Manuscript ID: sustainability-445695, with the title "Estimating Residential and Industrial City Gas Demand in the Republic of Korea-A Kalman Filter Application". In this paper, the author analyzes the city gas demand function in Korea from 1998 to 2016. I consider that the paper will benefit if the author addresses within the manuscript the following aspects:

Ø  The sections of the manuscript. Most of the sections of the manuscript in its actual form are according to the ones recommended by the Sustainability MDPI Journal's Template. However, the very important "Discussion" section, recommended by the Sustainability MDPI Journal's Template, is missing. The manuscript under review will benefit if the author devises an appropriate "Discussion" section and add it to the manuscript.

Ø  Line 11, the "Abstract" of the paper. "This paper attempts to estimate the city gas demand function in Korea from 1998 to 2016."  First of all, I consider that the expression "attempts to estimate" is not the best choice, maybe "analyzes", or "tackles issues related to" ("attempts to estimate" implies some uncertainty). Secondly, I consider that it will benefit the manuscript and in the same time it will highlight even more the author's contribution if he emphasizes more the purpose and usefulness of obtaining the city gas demand function in Korea from 1998 to 2016, today, in the year 2019.

Ø  Lines 11-19, the "Abstract" of the paper. Taking into consideration that this is a research article, the abstract should offer a relevant overview of the work. It will benefit the paper if the author provides a structured abstract, that covers the following aspects: the background (in which the author should place the issue that the manuscript addresses in a broad context and highlight the purpose of the study), the methods used to solve the identified issue (that should be briefly described), a summary of the article's main findings followed by the main conclusions or interpretations. In the abstract the author must also declare and briefly justify the novelty of his work. The author must state more clearly his purpose, his methods, his original results and conclusions as well as the novelty of his study. In the actual form of the manuscript, the abstract offers information related only to some of these aspects and even so, their delimitation is unclear.

Ø  Lines 22-80, the "Introduction" section. In this section, the author must introduce a presentation of the current state of the research field by reviewing it carefully and by citing key publications. By doing so, the problem will be put into context and it will benefit the readers as well. In the actual form of the manuscript, the state of art regarding the paper's field of research is too short. I consider that the article under review will benefit if the author extends this section by citing more publications from the multiple relevant ones that exist in the scientific literature regarding the article's topic. After having performed a critical survey of what has been done up to this point in the scientific literature, the author must identify a gap in the current state of knowledge that needs to be filled, a gap that is being addressed by his manuscript. This gap must also be used afterwards, in the "Discussion" section (which is currently missing from the paper) of the manuscript as well, where the author should justify why his approach fills the identified gap in rapport with previous studies from the literature. In the "Introduction" section the author must also declare the novel aspects of his work. At the end of the Introduction section, the author must present the structure of his paper under the form: "The rest of the paper is structured as follows: Section 2 contains…".

Ø  The "Materials and Methods" section. In order to help the readers better understand the methodology of the conducted research, the author should devise a flowchart at the beginning of the "Materials and Methods" section, a flowchart that depicts the steps that he has processed in developing his research and most important of all, the final target. This flowchart will facilitate the understanding of the proposed approach and in the same time will make the article more interesting to the readers if used as a graphical abstract.

Ø  The "Materials and Methods" section. It will benefit the paper to specify, in the final part of the "Materials and Methods" section, details regarding the version numbers for the software and the detailed hardware configuration used within the research.

Ø  The equations within the paper. All the equations within the manuscript should be explained, demonstrated or cited, as there are some equations that have not been introduced in the literature for the first time by the author and that are not cited.

Ø  Lines 267-288, the "References" section. According to the Sustainability MDPI Journal's Template, the references must be numbered in the order of their appearance in the text (including citations in tables and legends) and listed individually at the end of the manuscript. In the actual form of the paper, the references are ordered in an alphabetical order instead of the recommended one. Please renumber and reorder the references in the References section, according to the recommendations.

Ø  The "Discussion" section is missing. In order to validate the usefulness of his research, in the "Discussion" section (which is currently missing from the manuscript), the author should make a comparison between his approach from the manuscript and other similar ones that have been developed in the literature for the same or related purposes. I consider that the paper will benefit if the author makes a step further, beyond his analysis and provides an insight at the end of the "Discussion" section regarding what he considers to be, based on the obtained results, the most important, appropriate and concrete actions that the decisional factors and all the involved parties should take in order to benefit from the results of the research conducted within the manuscript as to attain the ultimate goal of sustainability.

Ø  Lines 87- 88, the "Data" subsection. "The consumption of residential and industrial city gas (in ) and price data (in Korean won, or “KRW”) were prepared using the Korean Energy Statistical Information System." The author must specify what was the data source by appropriately citing it.

Ø  Lines 83-85, the "Data" subsection. "This study estimated the demand function of Korea’s domestic and industrial city gas from January 1998 to July 2016 using its residential and industrial city gas consumption, price, and residential and industrial income." The authors must specify clearer in the manuscript (like I have mentioned in my comments regarding the Introduction section) what is the usefulness and impact today, in the year of 2019, of having computed the demand function of Korea’s domestic and industrial city gas from January 1998 to July 2016.

Ø  Lines 82-101, the "Data" subsection. The actual subsection "2.1 Data" must be completed with details regarding the data preprocessing. The author should specify what was his approach for the cases when data was inconsistent or incomplete, due to measurement errors. Can the author mention how much of his approach is being influenced by the used data or to which extent the approach can be easily applied to other situations, when the datasets are different?

Ø  The Figures from the paper. The axes' titles must be specified, along with the measuring units (if applicable).

Ø  The Format of the paper. There are unnecessary spaces between paragraphs. The author must take into account the recommendations regarding the format of the papers, by using the Microsoft Word template or LaTeX template to prepare his manuscript.

Author Response

Reply to the Review Report (Reviewer 3)

All mentioned comments are reflected in the attached revised paper with my best.

Please consider the positive and kind review for my revised paper.

Thank you for your honorable and substantial comments.

Comment

Reply

1.     The sections of the manuscript. Most of the sections of the   manuscript in its actual form are according to the ones recommended by the   Sustainability MDPI Journal's Template. However, the very important   "Discussion" section, recommended by the Sustainability MDPI   Journal's Template, is missing. The manuscript under review will benefit if   the author devises an appropriate "Discussion" section and add it   to the manuscript.

ü “Discussion” Section was added   in revised paper.

2.     Line 11, the "Abstract" of the paper. "This paper   attempts to estimate the city gas demand function in Korea from 1998 to   2016."  First of all, I consider   that the expression "attempts to estimate" is not the best choice,   maybe "analyzes", or "tackles issues related to"   ("attempts to estimate" implies some uncertainty). Secondly, I   consider that it will benefit the manuscript and in the same time it will   highlight even more the author's contribution if he emphasizes more the   purpose and usefulness of obtaining the city gas demand function in Korea   from 1998 to 2016, today, in the year 2019.

ü According to the comment, “attempts   to estimate” changed to “analyze” and, added contributions in revised papers

ü Period for data were extended   (1997-2018).

ü Re-estimation was done using the   changed data.

3.     Lines 11-19, the "Abstract" of the paper. Taking into   consideration that this is a research article, the abstract should offer a   relevant overview of the work. It will benefit the paper if the author   provides a structured abstract, that covers the following aspects: the   background (in which the author should place the issue that the manuscript   addresses in a broad context and highlight the purpose of the study), the   methods used to solve the identified issue (that should be briefly   described), a summary of the article's main findings followed by the main   conclusions or interpretations. In the abstract the author must also declare   and briefly justify the novelty of his work. The author must state more   clearly his purpose, his methods, his original results and conclusions as   well as the novelty of his study. In the actual form of the manuscript, the   abstract offers information related only to some of these aspects and even   so, their delimitation is unclear.

ü According to the comment,  added contributions and novelty in revised   papers with my best.

4.     Lines 22-80, the "Introduction" section. In this section,   the author must introduce a presentation of the current state of the research   field by reviewing it carefully and by citing key publications. By doing so,   the problem will be put into context and it will benefit the readers as well.   In the actual form of the manuscript, the state of art regarding the paper's   field of research is too short. I consider that the article under review will   benefit if the author extends this section by citing more publications from   the multiple relevant ones that exist in the scientific literature regarding   the article's topic. After having performed a critical survey of what has   been done up to this point in the scientific literature, the author must   identify a gap in the current state of knowledge that needs to be filled, a   gap that is being addressed by his manuscript. This gap must also be used   afterwards, in the "Discussion" section (which is currently missing   from the paper) of the manuscript as well, where the author should justify   why his approach fills the identified gap in rapport with previous studies   from the literature. In the "Introduction" section the author must   also declare the novel aspects of his work. At the end of the Introduction   section, the author must present the structure of his paper under the form:   "The rest of the paper is structured as follows: Section 2   contains…".

ü Additional literature studies   were done in revised paper.

§  Other Countries’ city gas or   natural gas demand-USA, China, Turkey, etc.

§  Other Energy   Industry-Electricity

§  Kalman Filter Method and   Applications

ü According to the comment,  added contributions and novelty in revised   papers with my best

5.     The "Materials and Methods" section. In order to help the   readers better understand the methodology of the conducted research, the   author should devise a flowchart at the beginning of the "Materials and   Methods" section, a flowchart that depicts the steps that he has   processed in developing his research and most important of all, the final   target. This flowchart will facilitate the understanding of the proposed   approach and in the same time will make the article more interesting to the   readers if used as a graphical abstract.

ü According to the comment,   introduced flowchart in revised paper

6.     The "Materials and Methods" section. It will benefit the   paper to specify, in the final part of the "Materials and Methods"   section, details regarding the version numbers for the software and the   detailed hardware configuration used within the research.

ü According to the comment,   specified the applied software in revised paper.

7.     The equations within the paper. All the equations within the   manuscript should be explained, demonstrated or cited, as there are some   equations that have not been introduced in the literature for the first time   by the author and that are not cited.

ü Citations for equations were   reflected in revised paper.

8.     Lines 267-288, the "References" section. According to the   Sustainability MDPI Journal's Template, the references must be numbered in   the order of their appearance in the text (including citations in tables and   legends) and listed individually at the end of the manuscript. In the actual   form of the paper, the references are ordered in an alphabetical order   instead of the recommended one. Please renumber and reorder the references in   the References section, according to the recommendations.

ü According to the Sustainability   MDPI Journal's recommandation, References are reordered.

9.     The "Discussion" section is missing. In order to validate   the usefulness of his research, in the "Discussion" section (which   is currently missing from the manuscript), the author should make a   comparison between his approach from the manuscript and other similar ones   that have been developed in the literature for the same or related purposes.   I consider that the paper will benefit if the author makes a step further,   beyond his analysis and provides an insight at the end of the   "Discussion" section regarding what he considers to be, based on   the obtained results, the most important, appropriate and concrete actions   that the decisional factors and all the involved parties should take in order   to benefit from the results of the research conducted within the manuscript   as to attain the ultimate goal of sustainability.

ü According to your honorable and substantial comments,   Discussion Section was written.

10. Lines 87- 88, the "Data" subsection. "The consumption   of residential and industrial city gas (in ㎥) and price data (in Korean won, or “KRW”) were   prepared using the Korean Energy Statistical Information System." The   author must specify what was the data source by appropriately citing it.

ü According to the comment, data   source was cited in revised paper.

11. Ø  Lines 83-85, the   "Data" subsection. "This study estimated the demand function   of Korea’s domestic and industrial city gas from January 1998 to July 2016   using its residential and industrial city gas consumption, price, and   residential and industrial income." The authors must specify clearer in   the manuscript (like I have mentioned in my comments regarding the   Introduction section) what is the usefulness and impact today, in the year of   2019, of having computed the demand function of Korea’s domestic and   industrial city gas from January 1998 to July 2016.

ü Period for data were extended   (1997-2018).

ü Re-estimation was done using the   changed data.

12. Lines 82-101, the "Data" subsection. The actual subsection   "2.1 Data" must be completed with details regarding the data   preprocessing. The author should specify what was his approach for the cases   when data was inconsistent or incomplete, due to measurement errors. Can the   author mention how much of his approach is being influenced by the used data   or to which extent the approach can be easily applied to other situations,   when the datasets are different?

ü According to the comment, details   regarding the data preprocessing was done in revised paper with cititation.

13. The Figures from the paper. The axes' titles must be specified, along   with the measuring units (if applicable).

ü According to the comment, axes’   title and unit were specified in revised paper.

14. The Format of the paper. There are unnecessary spaces between   paragraphs. The author must take into account the recommendations regarding   the format of the papers, by using the Microsoft Word template or LaTeX   template to prepare his manuscript.

ü According to the comment,   unnecessary spaces deleted.

Round 2

Reviewer 3 Report

After having reviewed the revised version of the paper "Estimating Residential and Industrial City Gas Demand in the Republic of Korea-A Kalman Filter Application", under the revised title "Estimating Residential and Industrial City Gas Demand Function in the Republic of Korea-A Kalman Filter Application", Manuscript ID: sustainability-445695, I have noticed that the author has addressed the most important signaled issues, therefore improving the manuscript.